Teaching computer architecture by designing and simulating processors from their bits and bytes

Doğan Mustafa 1 2 mustafadogan@aselsan.com.tr
http://orcid.org/0000-0003-2483-8070 Öztoprak Kasım 3
Tolun Mehmet Reşit 4
1 ASELSAN Research , Ankara , Turkey
2 Department of Computer Engineering, Hacettepe University , Ankara , Turkey
3 Department of Computer Engineering, Konya Food and Agriculture University , Konya , Turkey
4 Department of Software Engineering, Çankaya University , Ankara , Turkey
Leeser Miriam
Electronic publication date: 2024 Feb 19
Publication date: 2024
Volume: 10
Electronic Location ID: e1818
Received 2020 Nov 23; Accepted 2023 Dec 19
Copyright: © 2024 Doğan et al.
Copyright year: 2024
Copyright holder: Doğan et al.
License: This is an open access article distributed under the terms of the Creative Commons Attribution License, which permits unrestricted use, distribution, reproduction and adaptation in any medium and for any purpose provided that it is properly attributed. For attribution, the original author(s), title, publication source (PeerJ Computer Science) and either DOI or URL of the article must be cited.
License URL: https://creativecommons.org/licenses/by/4.0/

Keywords: Processor design, Processor simulator development, HDL implementation, Computer architecture, Integrated circuit, Hardware validation, Visualization

Funding: The authors received no funding for this work.

==============================
Teaching computer architecture (Comp-Arch) courses in undergraduate curricula is becoming more of a challenge as most students prefer software-oriented courses. In some computer science/engineering departments, Comp-Arch courses are offered without the lab component due to resource constraints and differing pedagogical priorities. This article demonstrates how students working in teams are motivated to study the Comp-Arch course and how instructors can increase student motivation and knowledge by taking advantage of hands-on practices. The teams are asked to design and implement a 16-bit MIPS-like processor with constraints as a specific instruction set, and limited data and instruction memory. Student projects include following three phases, namely, design, desktop simulator implementation, and verification using hardware description language (HDL). In the design phase, teams develop their Comp-Arch to implement specified instructions. A range of designs resulted, e.g., (a) a processor with extensive user-defined instructions resulting in longer cycle times (b) a processor with a minimal instruction set but with a faster clock cycle time. Next, teams developed a desktop simulator in any programming language to execute instructions on the architecture. Finally, students engage in Verilog Hardware Description Language (HDL) projects to simulate and verify the data-path designed during the initial phase. Student feedback and their current understanding of the project were collected through a questionnaire featuring varying Likert scale questions, some with a ten-point scale, and others with a five-point scale. Results of the survey show that the hands-on approach increases students’ motivation and knowledge in the Comp-Arch course, which is centered around computer system design principles. This approach can also be effectively extended to related courses, such as Microprocessor Design, which delves into the intricacies of creating and implementing microprocessors or central processing units (CPUs) at the hardware level. Furthermore, the present study demonstrates that interactions, specifically through peer reviews and public presentations, between students in each phase increases their knowledge and perspective on designing custom processors.

Introduction

In an era of rapid technological advancement, understanding the intricate workings of computer hardware, from the basic building blocks to complex architectures, is essential. Recent advancements in computer technology have led to increased complexity and abstraction, making it imperative for individuals to have a comprehensive understanding of computer architecture (Hennessy & Patterson, 2017). This need is particularly pronounced within the context of the Computer Architecture (Comp-Arch) course, an integral element of the Computer Engineering curriculum at Konya Food & Agriculture University (KFAU), and results in an intensified demand for individuals to possess an extensive comprehension of computer architecture, especially in educational contexts (Murdocca & Heuring, 1999).

Teaching Comp-Arch to undergraduate students presents a unique set of challenges (Theys & Troy, 2003; Kurniawan & Ichsan, 2017). In a world where many students gravitate towards software-oriented courses, the Comp-Arch course often appears as an insurmountable theoretical endeavor. Its demand for a profound comprehension of a computer’s internal mechanisms can be daunting to students. The shift of students towards software-oriented courses has been observed in various academic institutions, posing a challenge for educators to instill a deep understanding of computer hardware (Ristov, Stolikj & Ackovska, 2011). However, understanding the Comp-Arch course’s significance extends beyond academic achievement.

The Bureau of Labor Statistics (BLS) projects that jobs for computer hardware engineers will grow by about 2% from 2020 to 2030, which is lower than the average growth (8%) (U.S. Bureau of Labor Statistics, 2022a). However, under the same conditions, BLS projects jobs for information security analysts and web developers will grow by about 33% and 13%, respectively (U.S. Bureau of Labor Statistics, 2021, 2022b). Furthermore, BLS indicates that computer hardware engineers get paid relatively more than information security analysts and web developers (U.S. Bureau of Labor Statistics, 2021, 2022b, 2022a). This paradigm shift in career preferences not only affects the Comp-Arch course but also extends to other core courses in the curriculum. Within the KFAU Computer Engineering program, the first three semesters include common core courses. Subsequently, during the following five semesters, students delve into 16 essential courses, including Programming Languages, Computer Architecture, Operating Systems, Automata Theory and Formal Languages, Computer Networks where students often suffer from the problem of balancing their growing interest in software-oriented careers with the demand for understanding computer hardware (Noone & Mooney, 2018; Thomas et al., 2012; Anderson & Nguyen, 2005; Vijayalaskhmi & Karibasappa, 2012; Prvan & OžEGOVIć, 2020). As a significant portion of the curriculum, these theory-driven and low-level computer comprehension courses are poised to shape students’ career trajectories.

Despite these challenges, conventional pedagogical strategies have thus far failed to provide a practical and quantifiable approach to address these issues within the Comp-Arch course (Kurniawan & Ichsan, 2017; Theys & Troy, 2003; Nayak & Vijayalakshmi, 2013; Nova, Ferreira & Araújo, 2013). This study presents a series of hands-on assignments meticulously crafted to invigorate students’ enthusiasm for theoretical subjects and investigates the impact of these assignments on students’ motivation and their ability to share knowledge among peers. These assignments include: Processor design: Students are tasked with designing processors as depicted in Fig. 1, encompassing instruction set architecture (ISA) and data-path components (Nayak et al., 2021).

Desktop simulator: Students create a desktop simulator that illustrates how processors execute assembly code and monitors memory and register content changes during execution (Brox et al., 2018).

HDL verification: The designed processors are verified using hardware description language (HDL) (Velev, 2003).

Figure 1 Block-diagram of a simple CPU.

Upon completing these assignments, students are encouraged to share their thoughts and insights through a survey, providing valuable feedback on the course’s conduct. This approach not only enhances students’ motivation and understanding of theoretical subjects but also encourages them to explore diverse scenarios and use cases within their coursework.

The significance of this 14-week study lies in its pioneering approach to bolster students’ interest, motivation, and comprehension of the Comp-Arch course. First and foremost, this study presents a teaching model comprising the aforementioned assignments. This innovative teaching model offers a practical solution to a longstanding challenge and provides quantifiable results. Furthermore, it showcases the outcomes of student projects, with a multitude of teams each comprising 2–3 students. While a comprehensive presentation of all projects is beyond the scope of this study, we offer a compelling illustration of the remarkable progress achieved by showcasing two of these student projects. The study not only delves into the techniques employed by these students but also offers comprehensive documentation that is transparent and reproducible by fellow educators and researchers.

This innovative approach, which can be readily adapted to other courses, holds the potential to bridge the gap between theoretical knowledge and practical application. Moreover, it addresses the issue of low student engagement, especially in online or distance education settings (Sarder, 2014). Besides, if the requirements and constraints are arranged well, students can be more creative and focus on different aspects of user needs. By providing instructors with an effective assessment tool, it ensures that students receive the support and guidance needed to succeed in these challenging courses. Instructors can access and replicate this work shared on GitHub and YouTube, with small modifications to be used in their Comp-Arch classes (https://www.youtube.com/playlist?list=PL9PP7AO6lwHfsfOVjCUsG72CyZtLYKFtx).

The remainder of this article unfolds as follows: “Related Work” provides the background, while “The Processor (CPU) Design, The Simulator, and Verifying the CPU Design Using a Hardware Description Language” elucidate the design phase, simulator implementation phase, and hardware validation through Verilog. “Results” delves into the results, analyzing data collected from student surveys. The ensuing discussion in “Discussion” highlights the strengths and weaknesses of the project. Finally, the article concludes in “Conclusion and Future Work”, with a discussion of future directions and potential extensions of this research.

Related work

Many scientific studies have been conducted on the difficulties encountered in Comp-Arch and other theoretical courses. One of the best ways to tackle these difficulties is to give students effective hands-on assignments, such as simulators (Wu et al., 2014). In this section, studies are analyzed in three main parts: (a) courses that are hard to teach, i.e., computer architecture. (b) Possible problems and proposed solutions about why these courses are hard to teach. (c) Practicality and scope of simulators developed to teach Comp-Arch course.

Thomas et al. (2012) remark on difficulties and problems students can experience in a Comp-Arch course due to abstract concepts. Simkins & Decker (2016) and Omer, Farooq & Abid (2021) show that students grasp limited knowledge and the course itself becomes inefficient when teaching methods focus only on theoretical concepts (Simkins & Decker, 2016; Omer, Farooq & Abid, 2021; Yehezkel et al., 2001; Kehagias, 2016). Anderson & Nguyen (2005) survey the literature to find the best assignments for their course to avoid students struggling in the theoretical parts of an operating systems course. Vijayalaskhmi & Karibasappa (2012) state that teaching formal languages and automata theory courses is challenging due to the following reasons: (a) monotonous teaching style (b) courses mathematical nature causes poor understanding and students not showing adequate participation.

Leibovitch & Levin (2011) mention difficulties faced in Comp-Arch courses due to the fact that Comp-Arch courses are comprised of different fields, such as digital design, embedded systems, operating systems, assembly programming, instruction set architecture, instruction decoding, peripherals, etc. On the other hand, Patel & Patt (2019) state that the main issue is due to forcing students to memorize things before they understand the topic detailed. Simkins & Decker (2016) survey the difficulties that students encounter during programming courses. About 41% of students who encounter difficulties in “Tools for Learning” state that the main reason is lack of practice. Omer, Farooq & Abid (2021) collected and analyzed 66 different articles published from 2014 to 2020 to investigate recent developments in introductory programming course. Omer, Farooq & Abid (2021) and Malliarakis, Satratzemi & Xinogalos (2016) suggest using games to increase students’ motivation during the learning process. Furthermore, hands-on experiences with processor architectures have a supportive impact on students’ better understanding of the Comp-Arch course (Kehagias, 2016). In a comprehensive survey conducted by Kehagias (2016), every assignment, ranging from fundamental conceptual questions to 130 complex programming assignments, was meticulously analyzed within the context of Comp-Arch courses offered at leading North American universities. The conducted research examines the quality and quantity of assignments to enlighten the pathway for educators and instructors to create assignments and thereby assess students properly. Kehagias (2016) show that 25% of instructors include developing or modifying a simulator design task for a target processor architecture, which is a core part of this approach.

The necessity of hands-on experiences in teaching different courses is examined in several studies and various assignments or projects are proposed to contribute to students’ knowledge (Aviv et al., 2012; Hsu, 2015; Vijayalaskhmi & Karibasappa, 2012; Christopher, Procter & Anderson, 1993). According to a survey by Omer, Farooq & Abid (2021) survey tools are needed to have sufficient visualization to help students comprehend subjects. In their work, Morgan et al. (2021) developed the RISC-156 V Online Tutor, which not only incorporates significant interaction signals within real hardware, sandboxes, knowledge checks, and challenges but also provides a RISC-V Integrated Development Environment (IDE) for assembly programming, as well as VHDL processor capture, simulation, and prototyping. RISC-V IDE for assembly programming, and VHDL processor capture, simulation, and prototyping. Furthermore, the study by Wu et al. (2014) indicates that hands-on practices not only significantly improve students’ scores in an introduction to computer science course but also lead to a reduction in students’ stress levels during the course.

Nikolic et al. (2009) survey and evaluate the current simulators which are used to teach Comp-Arch course. The survey evaluates simulators in different categories which are the coverage of topics and features provided for simulation. The study shows that the best simulators that cover many topics are M5 and Simics simulators (Binkert et al., 2006; Magnusson et al., 2002). Several available simulators encompass approximately one-third of the Computer Architecture and Organization course content, focusing on fundamental aspects of Comp-Arch, memory system organization, architecture, interfacing, communication, device subsystems, and processor systems design. In contrast, M5 and Simics provide a more comprehensive coverage, spanning around two-thirds of the course material (Nikolic et al., 2009). Schuurman (2013) developed a simulator to teach processor architecture basics to computer science students. Schuurman (2013)’s approach shares common tasks with those advocated in this study, such as design and simulator phases. McGrew, Schonauer & Jamieson (2019) delve into the creation of RISC-V processors on FPGAs, accompanied by a comprehensive toolchain for simulation, assembly programming, and C-level code compilation. They aim to empower undergraduate computer engineering students with practical experience in CPU design, emphasizing a solid grasp of processor internals and compiler operations, while highlighting the relevance of the RISC-V ISA in real-world applications (McGrew, Schonauer & Jamieson, 2019). Vollmar & Sanderson (2005) introduce a Java-based MIPS assembly language simulator, specifically designed to cater to the needs of undergraduate computer science students and instructors, offering a user-friendly GUI and aligning with the straightforward design of the MIPS computer architecture commonly used in computer architecture and organization courses. Angelov & Lindenstruth (2009) designed a 16-bit RISC-based non-pipelined processor which can be created by entry-level students as course homework. Furthermore, they developed a simulator where users can type their assembly instructions and examine the code step by step. On the other hand, Bhagat & Bhandari (2018) did not only design a 16-bit RISC processor but also, verified their design by using Verilog HDL. Similar to Bhagat & Bhandari (2018), Angelov & Lindenstruth (2009) also used Von Neumann architecture. However, their design is limited to support 15 different instructions to make the processor simpler and easier to design. Jaumain et al. (2007) distinguished themselves in the realm of microprocessor education by introducing a simulation platform. This platform offers students the capability to input assembly instructions and meticulously track the progression of each electric signal in a step-by-step fashion (Jaumain et al., 2007).

In their study, Rao, Angeline & Bhaaskaran (2015) devised a processor design meticulously selecting components, including the ALU, control unit, and instruction decoder, with the goal of optimizing performance in terms of power efficiency and reduced delay. However, they achieved these results using 32-bit instructions while the ALU can perform 16-bit operations.

Black (2016) proposes a module to be used by students to allow them to run their designs in Arduino hardware with the help of an Emumaker86 simulator developed earlier by the professor. The study concentrates on allowing students to run their code in hardware rather than designing a processor. Similarly, Yildiz et al. (2018) introduced VerySimpleCPU (VSCPU), a soft CPU simulation platform, alongside the offered tool, to enable students to not only design their processors from scratch but also construct code for their processors, and seamlessly implement them using FPGAs, collectively enhancing the educational experience in computer architecture and organization courses at the undergraduate level.

The processor (cpu) design

In this section, we first describe the foundational knowledge essential for understanding the MIPS architecture. Then, it defines the fundamental design limitations which each processor must support. Finally, it analyzes and compares the differences of each designed processor.

Before delving into the intricacies of the processor design, it is essential to gain insight into the students’ educational background. The students who participated in this project possessed a strong foundation in mathematics, digital design, and basic Verilog knowledge, having completed prerequisite courses in programming (Java, Python, C/C++), discrete mathematics, and logic design. While their programming skills were adequate for the development of the desktop simulator, the questionnaire responses revealed that they encountered challenges during the design phase involving HDL, primarily due to their limited experience with HDL programming.

In this project phase, students were tasked with defining key elements of their processor designs, including instruction architecture, data-path, control signals, supported instruction list, and the design of the arithmetic logic unit. Each group had a 2-week period to brainstorm and conceptualize their architectural ideas. Following the principles highlighted by Omer, Farooq & Abid (2021) we fostered collaborative learning by having each group present their progress during various phases. During these presentations, students engaged in constructive discussions, providing feedback, and scrutinizing their peers’ designs. This interactive process allowed for valuable insights and improvements in the project as they could leverage each other’s strengths.

Initially, we had planned to introduce an additional phase focusing on the physical implementation of the students’ designs. We believed this phase would significantly enhance their understanding of the subject matter. However, the outbreak of the COVID-19 pandemic forced the university and its facilities to close during the semester, resulting in the unfortunate cancellation of this phase of the project.

To differentiate between the two distinct CPU architectures created by separate student groups, we will employ the abbreviations MuSe and DoMe.

MIPS architecture

This section provides a concise overview of the MIPS architecture. The Microprocessor without Interlocked Pipeline Stages (MIPS) architecture, a renowned example of a Reduced Instruction Set Computer (RISC) design, offers a straightforward and efficient 32-bit word size with a load-store architecture (MIPS, 2001). It features 32 general-purpose registers and supports diverse instruction types for various operations (MIPS, 2001). This simplicity and regularity make MIPS an excellent choice for educational purposes, allowing students to grasp computer architecture. We chose MIPS for its advantages in enhancing students’ understanding of computer architecture and organization.

When devising our CPU, we categorized instructions in a manner akin to the MIPS architecture, distinguishing between three primary types: R-type, I-type, and J-type instructions. Within the MIPS architecture, these instruction categories play a pivotal role, each with specific fields dedicated to fulfilling their unique functions (MIPS, 2001). The differentiation between these instruction types primarily hinges on their respective operation code (opcode) fields. Each instruction type possesses a distinct opcode value, except for R-type instructions. These R-type instructions, unlike their counterparts, do not encompass fields for target addresses, branch displacements, or immediate values. Instead, they feature fields for three registers, function codes, and shift amounts. The function code field plays a role in distinguishing R-type instructions from one another.

I-type instructions allocate the bits typically designated for the destination register, shift amount, and function code to represent immediate values. These immediate values serve various purposes, including acting as constant operands, branch target offsets, and memory operand displacements. I-type instructions offer users the convenience of incorporating constant values without necessitating a register. Meanwhile, J-type instructions facilitate alterations to program flow.

Prerequisites of CPUs

Since MIPS is a RISC type of architecture the students have limited instruction set, data and instruction memory, and register count. Each proposed architecture must support 18 predefined 16-bit instructions. These instructions are specified by the lecturer and given to students before the project starts. Students have separate 256-bytes program memory and data memory. Furthermore, they have eight 16-bit general purpose registers to access data that the CPU is currently processing. Each processor design must use a single cycle data-path and Von Neumann architecture to avoid complexity. With these specified instructions and constraints, architectures would be capable of writing many small-scale programs.

The CPU specifications were deliberately chosen to strike a balance between simplicity and comprehensiveness, leading us to opt for a 16-bit CPU design. While eight-bit CPU design was feasible, we excluded it due to its limited complexity, which might not align with the educational objectives of our course. On the other hand, the 32-bit CPU design was considered but ultimately dismissed because of the time constraints, as its development could not be completed within the designated time frame. These specifications were determined prior to the onset of the COVID-19 pandemic, with considerations for physical implementation that could potentially impact delivery schedules.

MuSe architecture: the processor design stage

This section provides a comprehensive overview of the MuSe Architecture, encompassing details on the Instruction Set, and Format. Furthermore, students delved into extensive work concerning ALU design, data-path development, and control signal configuration specific to their respective architectural projects. The MuSe Architecture is designed in such a way that it supports and conforms to all requirements mentioned in “Prerequisites of CPUs”-at the same time, taking care of performance issues which are encountered in the DoMe Architecture. “MuSe Architecture Design” contain a more detailed exposition of these aspects.

MuSe architecture: the instruction set and format

Our proposed instruction format, detailed in Table 1, places an emphasis on optimizing support for mandatory instructions. Consistent with the MIPS architecture, we have organized instructions into three primary types: R, I, and J. When selecting fields for our instruction formats, we prioritized the utilization of the three-bit opcode field as the ALU operation code, streamlining the instruction decoding process. This approach involves an initial assessment of the is_jump and is_imm fields to decode instructions efficiently. To minimize instruction decoding complexity, we sought to align three-bit opcodes with corresponding ALU operation codes, thus reducing the time and hardware resources required to determine the specific ALU operation. We adopted an alternative approach, adding supplementary fields known as is_imm and is_jump each consisting of one-bit length, to identify the instruction type. These added fields alleviate the need to rely solely on opcode distinctions, enhancing clarity in instruction classification.

Table 1 Instruction format of each architecture.

MuSe architecture	R-Type	Field	Opcode	is_jump	is_imm	rs	rt	rd	Unused	Total	
	Bit	3	1	1	3	3	3	2	16	
I-Type	Field	Opcode	is_jump	is_imm	rs	rt	Immediate		Total	
	Bit	3	1	1	3	3	5	16	
J-Type	Field	Opcode	is_jump	is_imm	Label	Total	
	Bit	3	1	1	11	16	
DoMe architecture	R-Type	Field	Opcode	control_bit	rt	rs	Function code	Total	
	Bit	4	1	3	3	5	16	
I-Type	Field	Opcode	control_bit	rt	Immediate	Total	
	Bit	4	1	3	8	16	

R-type instructions involve the use of three register values, while I-type instructions employ immediate values, omitting the (Rd) field in favor of a five-bit immediate field. J-format instructions, specialized for jumps, do not include the (Rd), Rs, or Rt fields. Instead, they incorporate an 11-bit label field, serving as the storage location for target addresses. These distinctions align with their intended functions, rendering the is_jump field set to 1 for jump instructions and the is_imm field invariably set to 1 for immediate instructions.

For a comprehensive overview of the various instruction types supported by the MuSe architecture, readers are encouraged to consult the “MuSe Architecture Design”.

DoMe architecture: the processor design stage

This section delves into the design principles of the DoMe architecture and highlights its divergences from the MuSe architecture. In contrast to the MuSe architecture, DoMe architecture places a strong emphasis on accommodating a broader range of instructions, extending beyond the mandatory set, including complex operations like division to enhance the CPU’s functionality. While this approach does introduce some variance from the MIPS philosophy, it broadens the processor’s capabilities, albeit with some associated performance trade-offs. Simultaneously, it affords students the opportunity to explore a wider spectrum of CPU functionalities and complexities.

DoMe architecture: the instruction set and format

In crafting the DoMe architecture, designers drew inspiration from the traditional MIPS architecture, just as the MuSe architecture did, thanks to MIPS’s reputation for simplicity and user-friendliness (MIPS, 2001). Table 1 provides an overview of the DoMe architecture’s instruction format, which exhibits several distinctions from the MuSe architecture in the following ways: (1) The DoMe architecture employs a five-bit function code. (2) It features an additional one-bit control_bit. (3) It omits the register address for the destination register Rd. (4) Utilizes an eight-bit immediate value compared to MuSe architecture’s five-bit value. (5) Dispenses with additional bits for is_imm and is_jump fields. (6) Forgoes the introduction of an extra instruction type for jump instructions.

In the R-type instruction format, the most significant four bits delineate the opcode, determining the instruction type. For R-type instructions, the CPU deciphers the instruction based on the function code. In other cases, the instruction is determined by the opcode, much like in the MIPS architecture. Following the one-bit specification for the resulting register, the next six bits serve to specify the source and target registers, each represented in three bits due to the eight general-purpose registers available. The least significant five bits designate the function code for identifying R-type instructions.

In the I-type instruction format, the first three fields align with those in the R-type format, except for the inclusion of an immediate field in place of source register and function code fields.

While DoMe architecture shares commonalities with the MIPS architecture, such as the presence of a function code, only two bits are allocated to the function code field, which may not sufficiently accommodate an extensive array of instructions under a single opcode. To promote diversity in projects, students’ approach is supported by the instructor, allowing students to propose reducing one of the register addresses in the instruction format to free up more bits in the function code field. DoMe architecture adopts a fixed, resulting register from among the eight general-purpose registers, instead of enabling the user to specify the register. Consequently, if a user seeks to perform an addition operation with the values of registers r1 and r2 and store the result in register r3, they must initially conduct the addition operation using registers r1 and r2. As a result, the CPU stores the outcome in a default register called (Rd). Subsequently, the user must transfer the resulting value from (Rd) to the intended register. Additionally, DoMe architecture introduces an alternative approach for result storage, granting users the ability to store the result in the target register using the control_bit specified in Fig. 2, allowing users to choose between these storage methods. If a user wishes to store the result in the destination register, the control_bit must be set to 1; otherwise, it should be set to 0. DoMe architecture designers append the “-c” suffix to their instruction set to differentiate where the result is stored. This distinction is exclusive to R-type instructions and does not apply to I-type instructions, as I-type instructions only employ a single register.

Figure 2 DoMe architecture control bit example.

The absence of a J-type instruction set is justified by the limited instruction memory of 256 Bytes in DoMe architecture. The I-type format of the DoMe architecture incorporates an eight-bit immediate part, facilitating access to any instruction within the instruction memory and enabling users to perform calculations with constants larger than 32 more conveniently. Further details on instructions in the DoMe architecture are available in the “DoMe Architecture Design”.

The simulator

Comp-Arch represents a multifaceted course that draws upon diverse elements from various domains of computer science, including operating systems and programming languages (Leibovitch & Levin, 2011). To optimize students’ learning experiences, this course typically incorporates practical lab sections in which students can apply theoretical knowledge (Nikolic et al., 2009). Many activities within the Comp-Arch lab can be effectively executed through simulators, making simulators an indispensable pedagogical tool for teaching Comp-Arch (Burch, 2002; Djordjevic, Nikolic & Milenkovic, 2005; Vollmar & Sanderson, 2006). As such, students are tasked with creating and presenting their simulators at different stages of the project, each of which must encompass the following essential features: (1) An interactive user input section. (2) The capacity to parse provided assembly code into machine code consistent with the processor’s design. (3) The ability to visualize the current values of registers and memory cells. (4) Options for code interpretation, including step-by-step execution and full automation. (5) Freedom to employ programming languages and tools of their choice.

Araujo et al. (2014) have suggested a visualization approach, MIPS X-Ray, as an alternative to full-system simulation. However, students are expected to create simulators resembling the exemplary simulator diagram depicted in Fig. 3. The user interface plays a pivotal role in controlling and visualizing the simulator, comprising five key units: (1) The Instruction Decoder is tasked with receiving and decoding assembly instructions that adhere to the students’ architecture through the user interface, with the decoded output seamlessly relayed to the Control Manager. (2) The Simulation Manager adeptly orchestrates the ALU and the Register and memory units, efficiently facilitating communication between these integral components. (3) The ALU, in turn, fulfills the crucial role of executing the requisite arithmetic calculations. (4) The Register and Memory unit plays its pivotal role by proficiently storing the contents of registers and memory, subsequently furnishing this vital data to the user interface. (5) The View Manager bears the responsibility of ensuring the timely and accurate updating of information stored within memory and register cells.

Figure 3 Example simulator diagram used for teaching Comp-Arch course.

Preceding the design phase, students were tasked with constructing a MIPS simulator compatible with 32-bit MIPS instructions, encompassing all features of the original MIPS processor (MIPS, 2001). This initial simulator served a dual purpose: firstly, to deepen students’ understanding of the MIPS architecture and provide a programming foundation for future simulators; and secondly, to grant students more time during the design and verification phases for developing desktop simulators tailored to their respective architectures. The choice of MIPS as the introductory architecture was underpinned by its innate simplicity, enhancing students’ ability to design processors from scratch. Students were allocated 3 weeks to complete their individual simulators.

Subsequent sections delve into the distinctions between various simulators and scrutinize the impact of public presentations on simulator design.

MuSe architecture: the simulator

In the development of the MuSe architecture simulator, the designers opted for the utilization of the Java programming language in conjunction with JavaFX for crafting an elegant graphical user interface. The simulator features a visually informative display of data memory, neatly presenting both the binary address and the associated value within that address in a user-friendly manner. To enhance comprehension, a dedicated section akin to the data memory is included for instruction memory, aiding in the visualization of machine code. Within the realm of the simulator’s register visualization, the current register names and their respective values are thoughtfully portrayed in a signed decimal format. An additional segment serves to display the status of control signals during the execution of instructions, with highlighted cues signaling their activation. Furthermore, an explicit representation of the current program counter is complemented by an LCD display that conveniently conveys crucial details encompassing the opcode, jump conditions, and immediate values of the ongoing instruction (Fig. 4). This inclusion of an LCD display within the simulator was inspired by its real-world counterpart used in physical implementations. Since the simulator already offers visualizations of registers, the students thought it prudent to incorporate the LCD Display section as a software tool, facilitating a detailed exposition of the instruction currently undergoing execution. Notably, both students and educators can readily employ this simulator as an insightful tool for the analysis and exploration of architectural design, allowing them to make modifications in accordance with their requirements (https://github.com/SevcanDogramaci/Processor-Simulator).

Figure 4 MuSe architecture desktop simulator screen.

For an in-depth exploration of MuSe Architecture’s simulator details, we invite readers to delve into the intricacies presented in “DoMe Architecture Simulator”.

DoMe architecture: the simulator

In the realm of simulator development, a distinctive approach was undertaken by the creators of the DoMe Simulator, where the primary emphasis was placed on the comprehensive representation of data residing within memory cells and registers, as vividly illustrated in Fig. 5. Diverging from the language choice of MuSe, the DoMe Simulator was crafted using the Python programming language for the architectural framework and leveraged PyQt5 for the graphical user interface. A notable feature that sets the DoMe Simulator apart is its dynamic visualization of both memory and stack functionalities, enabling users to closely inspect and monitor alterations in these two critical components simultaneously. The presentation of memory cell contents within the DoMe simulator distinguishes itself by offering not only binary representations but also decimal formats, which significantly eases the process of data analysis. Additionally, the DoMe architecture designers took the extra step of offering multiple data representation formats within the registers, encompassing hexadecimal, signed integer, and unsigned integer formats, catering to diverse user preferences. Nonetheless, it is worth noting that the DoMe simulator deviates from the MuSe simulator in terms of visualizing control signals and lacks the LCD display section discussed in the previous section. Both students and educators can harness the capabilities of this simulator not only for utilization and replication but also for actively contributing to this open-source endeavor by introducing novel functionalities (https://github.com/omer-metin/CPU16-Simulator).

Figure 5 DoMe architecture desktop simulator screen.

To gain a profound understanding of the intricacies behind DoMe architecture’s simulator details, we encourage readers to immerse themselves in the comprehensive insights provided in “DoMe Architecture Simulator”.

Verifying the cpu design using a hardware description language

In the realm of Comp-Arch education, students embarked on a journey of enlightenment, immersing themselves in the intricate concepts of Comp-Arch and the art of instruction handling. Their academic voyage culminated in the meticulous design of a processor that encompassed vital elements, including the CPU, ALU, instruction formats, and datapath, thoughtfully crafted to adhere to a designated instruction set architecture (ISA).

To illustrate their designs, students used a simulator to execute complex code sequences. The next step involved creating and simulating a Hardware Description Language (HDL) project that embodied their architectural concepts. While students had some flexibility, they leveraged conveniences provided by their integrated development environment (IDE), such as ‘for’ loops and basic arithmetic operations.

This phase lasted four weeks to accommodate the students’ foundational HDL knowledge. Most students favored using the Verilog hardware description language, and they implemented their HDL code in the Xilinx ISE Design Suite (Xilinx, 2007). Although Xilinx introduced the Vivado environment, students preferred ISE due to its lower system requirements.

Both student groups adopted the multiplication method proposed by Patterson & Hennessy (2016), which handled negative values by converting them to positive and deciding the sign bit later. While the MuSe Architecture designers implemented this method in Verilog, the DoMe Architecture designers chose to use the “*” operand to minimize simulation risks.

In the forthcoming sections, we will delineate the modules employed by each architecture.

MuSe architecture: the Verilog design

The Verilog design of the MuSe architecture, depicted in Fig. 6, is constructed using a set of seven distinct Verilog modules, each playing a crucial role in the overall functionality. These modules encompass: (a) The Instruction Memory module is responsible for housing instructions provided by the user in binary format, the Instruction Memory module takes the program counter as its input and returns the forthcoming instruction slated for execution. (b) The Data Memory module is a fundamental component that facilitates the management of data within the memory system. This module accepts data, addresses, and a suite of control signals as input and, in response, furnishes the data read from memory. (c) The Register File module is a repository of registers, this module empowers users to effect changes in the register set. It acts as the gateway for user-initiated alterations to the register contents. (d) The ALU module is the heart of the computation, the ALU module carries out a diverse array of operations, spanning addition, subtraction, multiplication, address calculations, and more. As outlined in the data-path structure, the ALU module receives two source inputs and a set of ALU control lines, using this information to execute the requisite computations and return the output value. (e) The Control Unit module is the pivotal module that takes as input the opcode, is_jump, and is_imm values, using them to dynamically update all associated control signals in a coordinated manner. (f) The Processor Testbench module serves as the central orchestrator, assuming the role of the primary module, orchestrating the functionality of the other modules, and ensuring seamless and harmonious operation. (g) The LCD module is an interface module thoughtfully designed to aid users in the inspection of register and memory cell contents. The LCD module enhances the user’s ability to visualize and comprehend the system’s inner workings.

Figure 6 Input and output of verilog modules in MuSe architecture.

The Verilog code for the MuSe Architecture is available on GitHub (https://github.com/SevcanDogramaci/Processor-Verilog-Simulation).

DoMe architecture: the Verilog design

DoMe architecture’s Verilog design shown in Fig. 7 utilizes Verilog modules. The followings are modules that DoMe architecture designers created for their HDL project: (a) Instruction Memory module contains instructions that are fed by a user in binary format. The module takes the program count as input and returns the instruction which is going to be executed. (b) The GPRs module describes eight general-purpose registers and stores the values. This module helps us to read and write in registers. (c) The Data Memory module contains an array with a length of 256 where each element represents one-byte of data just like in the simulator. As in the GPRs module, the Data Memory module provides us with reaching and changing the content of given memory cells. (d) ALU is a module where all the operations are done, such as summation, subtraction, multiplication, etc. As described in the data path, the ALU module takes two source inputs and one ALU control line array. According to these inputs, the ALU module carries out the necessary calculations and returns an output value. (e) The Control Unit updates the control signals and ALU operation codes for every instruction according to their opcode and function code. The updated control signals are used in other modules like Data Memory, GPRs, etc. (h) The RISC 16-Bit module is a container module to run the control unit module and data-path unit module together. These modules work simultaneously under the control of the RISC 16-Bit module.

Figure 7 Input and output of verilog modules in DoMe architecture.

The Verilog code for the DoMe Architecture is available on GitHub (https://github.com/mustafaadogan/RISC16-Verilog).

Results

In order to quantitatively assess students’ experiences and perceptions throughout the project, a questionnaire was administered, utilizing a ten-point Likert scale (Likert, 1932). The questionnaire consisted of a set of questions, the average, and standard deviation results are presented in Table 2. To mitigate order effects, the questions were presented in random order, and students were allotted a sufficient amount of time (1 h) to complete the questionnaire to avoid procedural bias. The questions were meticulously crafted to ensure clarity and neutrality, preventing any leading question bias.

Table 2 Questions and results from survey.

No	Question	Avg. Res.	Std. Dev.	
1	How would you rate your knowledge and experience level about hardware design before/after the course?	3.12/7.02	2.71/2.14	
2	How would you rate your self-learning ability before/after the course?	4.64/7.52	1.99/1.93	
3	How interested were you in working on low-level systems, such as Computer Hardware Engineering before/after the course?	3.92/4.92	2.50/2.89	
4	How aware were you of developing software by considering hardware abilities and limitations before the course?	6.98	2.13	
5	How did the use of visual tools during the course impact your learning experience?	3.84	1.04	
6	How did working in groups during the course affect your learning experience?	3.46	1.20	
7	How did hands-on experiences during the course influence your learning experience?	4.12	0.99	
8	How did implementing and designing your own architecture during the course affect your learning experience?	3.78	1.02	
9	How did using simulators and tools shared on the internet during the course impact your learning experience?	4.04	0.82	
10	How satisfied were you with the Computer Architecture course?	3.86	1.00	
11	To what extent did this course meet your expectations?	3.68	1.00	
12	How confident do you feel in your understanding of hardware design after completing this course?	3.96	0.84	
13	How well do you think the course prepared you for low-level system work, such as Computer Hardware Engineering?	3.88	1.19	
14	How would you describe your overall enjoyment of this course?	4.00	0.77	

The table displays the mean responses and standard deviations for each question, shedding light on participants’ experiences and perceptions both before and after the course. While the Likert scale questions (Questions 1–4) are rated on a scale from 0 to 10, the Likert-type questions (Questions 5–14) employ a five-point scale. It is worth noting that the Likert-type questions feature varied answer options; for instance, Questions 5–9 span from “Significantly Hindered (1)” to “Significantly Improved (5)”. A more detailed analysis of all questions and answers is available in the “Survey”. This table offers a comprehensive examination of the course’s impact on participants’ knowledge, skills, interests, satisfaction, and their inclination to pursue future opportunities in the fields of computer architecture and low-level systems.

The survey results reveal substantial improvements in students’ knowledge, skills, and interests following the Computer Architecture course. Notably, students’ self-reported knowledge and experience in hardware design demonstrated a significant increase, with the average score rising from 3.12 before the course to 7.02 after the course. Self-learning ability also saw remarkable enhancement, improving from an average of 4.64 to 7.52. Interest in low-level systems and Computer Hardware Engineering displayed a more moderate yet positive increase from 3.92 to 4.92. The course had a notable impact on participants’ awareness of software development with consideration of hardware abilities and limitations. However, some questions received more mixed responses, such as the effect of group work, hands-on experiences, and the use of simulators and internet tools. These results suggest that the course was highly effective in increasing students’ hardware design knowledge and self-learning ability, while also fostering an increased interest in low-level systems. While participants’ overall satisfaction, confidence in understanding hardware design, and preparation for low-level system work were slightly improved, they generally expressed a high level of enjoyment with the course.

The interest in low-level systems and computer hardware engineering showed a slight improvement rather than a significant one, which can be attributed to the variability in participants’ responses, as indicated by the relatively higher standard deviation for these questions. Although the average ratings increased from 3.92 to 4.92, the larger standard deviations (2.50 to 2.89) signify greater divergence in students’ responses. While some students experienced a substantial boost in their interest, others may have had more marginal changes or even a decline, resulting in an overall moderate increase in the average scores. This variance in individual responses could be due to various factors, including participants’ prior interests, personal goals, and experiences, which influenced their perceptions of the course’s impact on their interest in low-level systems and computer hardware engineering.

In addition to the questionnaire analysis, a 1-h meeting was conducted among students and instructors to delve into the results and gather further insights. Several noteworthy points emerged from this discussion: (a) Sequencing the implementation of the MIPS simulator before embarking on the CPU design phase significantly boosted students’ confidence and awareness during the latter stage. (b) Given that the project combines both programming and electronic skills, students exhibited a preference for collaborative group work over individual efforts. (c) Participants observed that engaging in discussions and presentations at each phase of the project enhanced their understanding of the topics, even though this approach entailed additional workload. (d) Due to COVID-19 constraints, students were unable to complete the physical implementation component, but they expressed a belief that conducting designs on an FPGA board instead of using ICs and breadboards would enhance their grasp of embedded systems. Additionally, with university laboratories equipped with ample FPGA boards, the physical implementation phase would be cost-effective compared to TTL implementation. (e) Students attributed the enhancement of their design to their supervisor’s meticulous guidance and consistent advice provided during presentations and discussions, resulting in a more foolproof project.

Discussion

The Comp-Arch course offers a comprehensive curriculum, which can be challenging to cover in a mere 14-week duration. The three-phase processor design project successfully addresses the fundamental components of the course while providing students with a tangible experience. Through the use of discussions, peer reviews, presentations, and interviews during the project, abstract concepts are transformed into concrete knowledge. The course’s flexibility allows instructors to tailor the scope to individual students’ backgrounds and capabilities, potentially enhancing their learning experience. Nevertheless, to make the most of this course, students should ideally possess intermediate-level skills in programming, logic design, and HDL development.

The results of the post-project questionnaire align with previous observations and related research. Students emphasize the significance of the simulator (average rating of 4.04 out of 5) and the processor design phases (average rating of 3.78 out of 5). Simulators contribute to students’ theoretical knowledge, while the processor design phase compels them to explore the functionalities of core components in the Von Neumann architecture, including the control unit and the logic unit. While the HDL implementation adds a fresh perspective to students’ learning, it is important to acknowledge the substantial effort involved in this approach. Students also express concerns about the workload associated with this method.

Prior research often focused on either designing high-performance CPUs or creating CPUs and tools for educational purposes. Many studies revolved around demonstrating, implementing, and evaluating CPU designs from scratch. Additionally, various surveys targeted different courses, including Comp-Arch, to assess students’ opinions and potential improvements to ongoing challenges. These surveys underscore the value of hands-on practical experience in theoretical courses. Instructors teaching the Comp-Arch course can readily adopt this approach, and students can access publicly available documents and videos. The approach’s adaptability is one of its strengths and can be applied to various fields.

Despite our efforts to enhance the teaching of the Comp-Arch course, our study does not include a physical implementation phase. The physical implementation of designs in a lab setting offers valuable hands-on experience, enabling students to statistically evaluate their design’s performance. Unfortunately, the COVID-19 pandemic prevented students from completing this phase, which can be considered for future enhancements. An FPGA implementation could be integrated as a final assignment to provide students with an opportunity to expand their practical workload.

Our study highlights that students engaged in hardware design and implementation often possess a foundational understanding of integrating software with hardware, taking into account the capabilities and limitations of the hardware, even before enrolling in the course. Simulating the assembly language instructions improves their comprehension of hardware and software interfaces, which will be invaluable in constructing intricate computational systems. Moreover, our results indicate that the three-phase processor design project has a slightly positive influence on students’ career choices.

Conclusion and future work

The evolving landscape of computer science has seen students increasingly drawn to programming and related areas, despite the generally lower compensation in these fields compared to hardware engineering. Additionally, the complexity of certain courses poses challenges, as some students may struggle to grasp the material fully, potentially impacting their career prospects. As a response, this study delves into the effectiveness of a hands-on, team-oriented approach in motivating undergraduate students to engage with Comp-Arch, bolstering their knowledge in this domain. This approach particularly emphasizes the design and implementation of 16-bit processors across three project phases, evaluating its impact through a Likert scale-based questionnaire. In essence, our proposal introduces a three-phase processor design project for integration into Comp-Arch courses, with the aim of making it accessible to instructors and facilitating potential enhancements. This approach seamlessly blends creative thinking and hands-on practice while incorporating peer reviewing and public presentations. It can serve as a foundational model for other computer engineering courses that require a solid grasp of low-level computer understanding. This article offers insight into students’ processor design experiences and highlights the knowledge diffusion that occurs within teams as they concentrate on different CPU aspects. The study underscores that, with careful guidance, students can successfully design fully functional processors and their corresponding assembly languages. The gain of the present study can be summarized as: “Comp-Arch course would be very interesting and beneficial when proper tools and assignments are provided.”

In terms of future work, this study opens avenues for scientific exploration and educational enhancements. Researchers can explore the scope of this approach within the Comp-Arch course, suggesting alternate approaches to cover diverse content, such as pipelining, while maintaining student motivation. Addressing potential tool-related difficulties, as noted by Omer, Farooq & Abid (2021), researchers can introduce more user-friendly tools and assess their effects on students’ development, as exemplified by RISC-V Online Tutor (Morgan et al., 2021). Furthermore, the study acknowledges the high workload placed on students by this approach. Future investigations could break down the effect of each project phase, quantifying the workload-to-benefit ratio. This insight can guide educators in achieving comparable results with reduced student workload. Finally, considering students’ keen interest in FPGA board-based designs to enhance their understanding of embedded systems, a promising avenue for future work could involve integrating the further development and evaluation of this FPGA-based implementation approach as an additional phase within the existing three-phase project structure. Leveraging university resources with ample FPGA boards, this approach could prove to be cost-effective and align well with contemporary advancements in embedded systems education.

Supplemental Information

Supplemental Information 1 Architecture Details, Simulator Details, and Survey.

The authors express their gratitude to Ömer Metin, Mustafa Çataltas, and Sevcan Dogramaci for their valuable contributions to the design and reporting processes (Dogan, 2023; Metin & Dogan, 2023; Dogramaci & Cataltas, 2023a, 2023b).

Additional Information and Declarations

Competing Interests

Author Contributions

Data Availability

Mustafa Doğan is employed by Aselsan Research.

Mustafa Doğan performed the experiments, analyzed the data, performed the computation work, prepared figures and/or tables, authored or reviewed drafts of the article, and approved the final draft.

Kasım Öztoprak conceived and designed the experiments, authored or reviewed drafts of the article, and approved the final draft.

Mehmet Reşit Tolun analyzed the data, authored or reviewed drafts of the article, and approved the final draft.

The following information was supplied regarding data availability:

The codes and videos are available on GitHub, YouTube, and Zenodo. There are simulator codes in Java and Python, and Verilog codes.

- MuSe Simulator, Processor-Simulator, https://github.com/SevcanDogramaci/Processor-Simulator.

- Dogramaci, S., & Cataltas, M. (2023). MuSe Processor Verilog Codes. Zenodo. https://doi.org/10.5281/zenodo.10204853.

- DoMe Simulator, CPU16-Simulator,

https://github.com/omer-metin/CPU16-Simulator.

- Metin, O., & Dogan, M. (2023). DoMe CPU16 Simulator Codes. Zenodo. https://doi.org/10.5281/zenodo.10204824.

- MuSe Verilog, Processor-Verilog-Simulation, https://github.com/SevcanDogramaci/Processor-Verilog-Simulation.

- Dogramaci, S., & Cataltas, M. (2023). MuSe Processor Simulator Codes. Zenodo. https://doi.org/10.5281/zenodo.10204841.

- DoMe Verilog, RISC16-Verilog, https://github.com/mustafaadogan/RISC16-Verilog.

- Dogan, M. (2023). DoMe RISC16 Verilog Codes. Zenodo. https://doi.org/10.5281/zenodo.10204805.

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
