# Peer review of "Teaching computer architecture by designing and simulating processors from their bits and bytes"

_PeerJ Computer Science, doi:10.7717/peerj-cs.1818_

## Round 0.1 · original submission · Major Revisions

Please pay attention to the reviewers' detailed comments when submitting your revised manuscript.

Reviewer 1 ·

Basic reporting

- The English writing is okay, but there are a number of errors. For example: "The project comprised four phases" should be "The project is comprised of four phases".
- Similarly, there is some awkward phrasing in the document that should be improved. For example, "prefer to indulge in more" would read better with something like "prefer". I don't want to edit the entire document, but in the abstract, I count approximately 15 of either awkward phrasing or errors. This seems too high and should be improved.
- "fade out the users’" - I think the authors mean that abstraction and complexity make it harder for users to understand the low-level details of the machine.
- Change citations to authors - To eliminate this bias, Schuuman [1] developed a simulator to...
- The background research in this area is placed in the introduction section. It does not read well with the introduction and should be put into another section.
- I think there are some citations missing and these 3 citations should be included and used to find a number of other missing citations including in bibtex form: @book{patel2019introduction, title={Introduction to Computing Systems: from bits \& gates to C \& beyond}, author={Patel, S and Patt, Yale}, year={2019}, publisher={McGraw-Hill Professional}}; ---- @article{nikolic2009survey, title={A survey and evaluation of simulators suitable for teaching courses in computer architecture and organization}, author={Nikolic, Bosko and Radivojevic, Zaharije and Djordjevic, Jovan and Milutinovic, Veljko}, journal={IEEE Transactions on Education}, volume={52}, number={4}, pages={449--458}, year={2009}, publisher={IEEE}}; -----@inproceedings{mcgrew2019framework,
title={Framework and Tools for Undergraduates Designing RISC-V Processors on an FPGA in Computer Architecture Education}, author={McGrew, Tyler and Schonauer, Eric}, booktitle={2019 International Conference on Computational Science and Computational Intelligence (CSCI)}, pages={778--781}, year={2019}, organization={IEEE}};
- The article "the" is used too much when "a or an" is more appropriate.
- Do not see a clear hypothesis stated or research question

Experimental design

- Lacks strong experimental evaluation of the results (I provide suggestions in my final comments). I believe this is a case study of an intervention, but no clear question was answered nor was a method of the evaluation presented.

Validity of the findings

- Lines 456-461: Students have been observed in previous studies. The question is, what value do we get from 2 groups and what can we say about the differences.
- The following: "The study's gains can be summarized as: “designing a processor is as easy as solving high school math problems”." seems a little overzealous. Students need to be familiar with programming, HDL design, and other tool usages that seem to have been left out as background knowledge. I doubt a 2nd-year university student could build an ALU in Verilog, let alone build the datapath and control. I think more context needs to be provided on the state of the student when entering this experience. As a senior CpE or CS student with a background, this project seems doable. This, however, means that the student has spent 4 years of learning beyond the "High-school" stage. I would agree students can build processors, but your statement seems misleading.
- Lines 491: How does this show that they were interested in assembly programming? The projects were in architecture design and emulation. Not assembly programming.
- Line 493: "easily" = I disagree that this conclusion can be made based on the presentation.

Overall, I would argue that the case study provides little in the way of findings.

Additional comments

Overall Comments:
This paper provides us with two contrained MIPs architecture designs for a course via ISA implementation, emulation/simulation, HDL design, and chip-based realization (though the last step was not completed due to Covid-19). Two groups implement their respective design through 3 of the 4 stages and are compared for differences in choices. A discussion section, briefly, describes how the two groups experiences were.

As explained in my comments, I don't see a strong educational impact of the results. I believe the exercise was good, but I do not see how another educator can use this knowledge to determine: 1) if they should take this approach and 2) how this approach will improve the education of learning outcome(s). The aspect I was most interested in was the simulator design, which provided very little details on how I might take this approach and include it in my classes. Where I think this work should proceed is for the authors to repeat the experiment, and try to answer my questions 1 or 2 (or both) by implementing a more detailed experiment. The post interview is fine, but I might inlcude a pre and post test to see if knowledge has increased. Maybe a Likert scale to get some quantitative results. Otherwise, this paper is a fine documentation of a case intervention that seems to have gone well; however, there is little I can directly use from it.

Other comments:
- Verilog HDL and VHDL are, arguably, equivalent from a synthesis standpoint. The statement "However, they carried out their studies up to the VHDL level." makes no sense.
- What is a "proper documentation methodology" (line 126)? I have never heard of such a thing in educational research and would expect it to be cited.
- Lines 131-134 should be rewritten more simply as it is just describing what is to come.
- Lines 153-155: It is not clear why 16-bit was chosen. There's no reason 32-bit would be harder that I can come up with, so what are "some difficulties"?
- Section 2.2: This is ISA design choices. The terms R, I, and J should be defined for the reader and not used as if they know them.
- How did the students come up with adding the syscall instruction. Also, what does that instruction mean in a hardware form? I can understand that both SPIM and MARs emulators have such an instruction, but does it have meaning at the hardware level?
- Section 2.2.3: A multiplier can be implemented in many different ways. Since this is implemented in Verilog, I would expect the "*" instruction to be used and either soft-logic or a hard-multiplier being used. An iterative algorithm is risky, and how does it deal with negative values? This is probably the most important part of multiplication. Division is probably the real challenging ALU component.
- Section 2.3: Lines 324-239 - this seems like methodology and group discussions. Not sure if it fits with the description of the 2 groups processors.
- Lines 290-298: It seems like different fonts are being used. This ruins consistency.
- Section 2.3.3: Wallace tree multiplication should be cited.
- Section 2.3.3: Is Wallace tree costly to implement? Do you mean with TTL chips or on an FPGA?
- Line 328: "MISP"...
- Section 3.1 and 3.2: These two sections are glossed over a little too much. Are these emulators or simulators? How are they designed? Is there a loop to simulate the fetch, decode, execute, writeback phases? Is memory just a simple array? This aspect of the paper is the most interesting since this is an approach I have rarely seen before. It would be interesting to know the starting assignments difference from the end delivery of the simulator.
- Section 4: Nothing is mentioned on what skills these students already have. It seems they are already familiar with Programming and HDL design? They must also be familiar with TTL logic and breadboarding.
- Section 5: Not sure why the hardware simulation has to be done for TTL implementation. It seems that at the HDL point, the typical implementation would be mapped to an FPGA in modern electronic design. Going to TTL is interesting, and likely, a factor of available hardware. Also, there are no images of the hardware. My guess was that the exercise was just selecting components that could be put together to make a realized hardware design. Again, I would argue the FPGA implementation is a better educational time for these students.
- Discussion: Details of the interview process should be specified. How long were the interviews? How was the "feelings" interpreted/encoded from the interviews? How was researcher bias dealt with when we have no quantitative data?
- Future work: I would be more interested in what the educational future work is and not how the students can improve their processors. Pipeline implementation would be an interesting problem, but why? What educational benefit will these students get from that exercise?
- Figure 3 and Figure 4 are directly from Paterson and Hennessy's book and should be acknowledged correctly.
- Figures and Tables in general should add to our understanding. In this case, they seem mostly showing the two designs. They're not very informative.

·

Basic reporting

The paper describes the practical elements of an interesting course in computer architecture.Tasks include processor instruction and architecture design, HDL capture and HDL simulation, development of an assembly program simulator, assembly/test using discrete components (though limited detail) on the latter.

I like the idea of the paper, since I work in the area myself and recognise the need for case study publications.

The paper describes the steps and achievements in comp arch course practical work.
Such papers can be very useful in supporting and encouraging other professors, with practical details of experience.

A review of paper clarity and grammar is necessary (comments in attached pdf). Improved alignment between diagrams and text would improve the paper.

Consider including some further related references

Structure presents each element of the practical work. A review would improve the structure and clarity (refer to comments pdf)

Results. Include a reference to a repository with example achievements and course specifications if possible, though bear in mind that this could pose an issue with future students having access to solutions.

Experimental design

The paper provides a case study rather than fundamental research.

A reasonable level of detail on the experiments (labs) is included.
Recommend reviewing figures and accompanying text to tighten up the links and shorten the text and/repetitionwhere possible. A thorough review of structure, grammar and clarity would benefit the paper.

Computer architecture practical work is core to high quality coursework. The paper contribtes to this.
I am not entirely sure if J Comp Science is the best paper for this work, since processor design, HDL model capture, simulation is not common in comp science courses.
Another could be the Journal of Engineering Education.

Improved feedback data would enhance the paper.

Validity of the findings

The paper is useful to professors who wish to replicate this work, particularly is a repository is provided.

A review of conclusions and review/alignments of abstract/intro and conclusions/future work would enhance the paper.

Additional comments

I like this paper.

It is an interesting case study of pratical coursework.
The paper would benefit from a thorough review. I have indicated acceptance with minor revisions, though a significant review of structure and clarity would benefit the paper.

---

## Round 0.2 · Major Revisions

Be sure to address the concerns of the reviewers when preparing your revision.

Reviewer 1 ·

Basic reporting

The text has improved significantly, but there are still some problems that can be fixed.

Experimental design

It is appropriate in the education space but is somewhat light.

Validity of the findings

The findings are more interesting and improved over the previous submission. Still, I believe the authors need to evaluate the results in terms of letting the reader know that this is their interpretation of the survey data.

Additional comments

Overall, the paper has improved and is approaching a valuable contribution, but I ss to be refined. Here are my key points:

- The Likert scale results are a definite improvement, but the questions seem to get to the real questions in a roundabout way. Consider questions that are more direct "Did you enjoy this experience?". I will accept the approach, but the authors should note that the results are interpretations of the data.
- I don't see an online release of this. This is promised in the rebuttal.
- The writing can still be improved...I mention a few below, but I did not track every problem.
- Some of the figures are artifacts of the process but have no value in the journal setting. For example, why do I need to see a waveform of the verification process, when that image brings nothing to the article's purpose?

Overall, I appreciate the author's improvement of the work and hope that the next attempt will improve it even more. Note that some of the images (even though cited) might need to be considered for permission to use as they are directly from other sources.

Comments:
- "Firstly," implies a secondly...
- Still missing some citations in the background:
Simulators - http://courses.missouristate.edu/kenvollmar/mars/ for the Mars emulator
CA HW implementations - Framework and Tools for Undergraduates Designing RISC-V Processors on an FPGA in Computer Architecture Education. P. Jamieson, Tyler McGrew, and Eric Schonauer. 6th Annual Conf. on Computational Science & Computational Intelligence (CSCI'19)
- Not sure about the relevance of the multiplier discussion
- "there left only two bits" typo
- Figure 3 does not provide much value, and the caption that states Design is incorrect. This is just a block.
- All the images will require copyright releases. This might be concerning.
- Figures 10 and 11 do not add any value. Yes, the tool creates waveforms for simulation.
- Table 2 - I would argue that "." is more appropriate than "," for the Journals target audience

·

Basic reporting

Article is stil a difficult read. Needs further grammatical and clarity review.
Good background section references.
Figures could be improved, adding more detail (comments in pdf)
Case studies presented
Survey presented in this revision, though before/after numbers do not indicate impressive improvements in student learning etc. This needs some further consideration. Also some survey questions are difficult to undrstand, and some seem to be missing from the survey table (Table 2)

Experimental design

Overcoming the challenges in effective teaching of computer architectures, design to hardware prototyping is stated though is a common challenge.
Technial details of student MIPS architecture submission is lengthy and tedious in places. Clarity could be further improved (see df comments)
Could include a quick summary or reference to main MIPs architecture elements and instruction set early in the paper, which may simplify and shorten later sections on studnet submissions.

Validity of the findings

Differences in student project submission are pointed out, though too much detail in places.

Seems that 2 goups of 10/11 students submit work (Since 21 survey respondents and two architectures submitted. This is my interpretation. I was not aware of this in the first review. These are very large groups, whose size and individual contributions require management. Engagement etc issues could impact survey responses.

Include individual survey details (charts), beyond only providing averages.

Additional comments

Full comments:

The reporting of teaching computer architecture is interesting. Striving to develop and measure effective teaching methodologies is to be encoraged. So I complement the author(s) on their efforts.

The goals of the work, as I understand it, are to provide effective and measurable student learning through including group-based activities on
- computer architecture (comp arch) design
- simulator development
- HDL capture, simulation (and possibly FPGA hardware implementation, not possible due to Covid)
- presentation and peer review

The article highlights (section 2) previous work in reporting challenges in teaching computer architecture.
I believe that many professors successfully provide similar course, though may not have submitted their detail to publication review. The authors need to more clearly capture how their work makes a significant contribution.

The article is still a difficult read.
I have included lots of hand-written comments in the pdf, too many to list here.
Grammar issues and spelling errors (pointed out in pdf) still exist.
The article quality and clarity must be high to be considered for publication.

While it describes the hands-on steps in the course and differences in project submissions, it does not strongly measure the effectiveness of the approach in survey results (no included). This needs more consideration.

Although this may seem severe, I recommend that the principles could be refined and re-applied in a 2022/2023 semester, following reviewers' feedback, with smaller groups, improved clarity on student submissions, and above all with a careful review for article clarity, survey questions and analysis, aiming at obtaining more clear outcomes, recommmendation and conclusions.
I encourage the authors to continue to refine their efforts, even if this deays pubication.

The article describes the detail of two architectures, submitted by students.
The article provides survey feedback from 21 students. Two architecture subissions are mentioned, MuSE and DoME.

Were there two groups of 10/11 students? This is not clear (mentions that there were 21 students in the survey). If large groups performed the tasks, provide details on how work was managed within the groups. Did all students contribute? If not, this would impact survey results.

The aritcle would be more clear if the descriptions of the two submitted architectures was more clear and shortened, The text description of the two architectures is too detailed.
Consider
- including an early section describing the main aspects of MIPS architecture and intruction set, or providing a good reference.
- reducing the need to spell out MIPS arch details. Rather focus on the students' solution only.
I got bogged down in reading the details at times.
Descriptions would be more clea rand shorter too if more complete labelled diagrams of the architecure are included.

Present survey results (Table 2) more clearly (see comments in pdf)
Number the survey points.
Some survey results, e.g, effect on student career choice ar enot in the table, so it is not posible to understand the points made in the text.
Survey results before and after are not significantly different. This raises questions about the effectiveness of the approaches. Why are 'after' results not higher. Needs discussions.
Number the survey questions. Consider including chart of all answers, not only averages.
Survey questions "if you were not exposed to" are not clear, sice there is no 'if you wer exposed to' option.

I recommend replacitng CA with CompArch or similar throughout the article, since CA tpyically refers to continuous assessment.

---

## Round 0.3 · accepted · Accept

Please consider the recommended minor revisions when preparing your final article.

Reviewer 1 ·

Basic reporting

I believe the paper is in a reasonable space from my perspective. I'm not convinced of a major contribution, but the materials added to the space by the authors will have some benefit to future educators.

Experimental design

no comment

Validity of the findings

no comment

Additional comments

Still a few typos and misworded sections. One more review should complete this.

·

Basic reporting

The paper structure and readability has significantly improved.
Survey results indicate good outcomes for students, having followed the course.

Experimental design

The paper highlights a series of related practical elements in a computer architecture course, followed by collection and analysis of student feedback.

Feedback indicates good success and student satisfaction.

Validity of the findings

Well presented results and discussion.

Additional comments

Some specific comments with line numbers

134 "instruction set structure, deciding"
Should this be "instruction set architecture, instruction decoding, peripherals"?
Consuder adding memory architecture.

133 "different contiguous fields" Not sure if this is correct


158 In their work, Morgan
159 et al., 2021 developed the RISC-156 V Online Tutor, which not only incorporates significant
160 interaction signals within real hardware, sandboxes, knowledge checks, and challenges but also
161 provides a RISC-V Integrated Development Environment (IDE) for assembly programming, as
162 well as VHDL processor capture, simulation, and prototyping (Morgan et al., 2021). RISC-V
163 IDE for assembly programming, and VHDL processor capture, simulation, and prototyping

I mentioned this (my own) reference in an earlier review as an example of practical hands-on learning of computer architecture, since references to existing hands-on practice with hardware were missing.
A better approach may be to include a few other hands-on practice references, e.g, LabsLand online platform.

If using the reference, review (shorten) the reference wording (162 sentence (includes repetition))
In their work, Morgan et al., 2021 developed the RISC-V Online Tutor, which incorporates interaction with real FPGA hardware signals, sandboxes, knowledge checks, and challenges, and a RISC-V assembly programming Integrated Development Environment (IDE).(Morgan et al., 2021).


Other simulators (RISC-V) https://www.riscfive.com/risc-v-simulators/


203-206 repetition


268 Students have separate 256-bytes program memory
This seems very small, not supporting many instructions (32 max?)


622-623 Comp-Arch course would be very interesting and benefits when proper tools and assignments are provided
Comp-Arch course would be very interesting and beneficial when proper tools and assignments are provided

Figure 12: should this graph include data for before and after the course? It suggests strong knowledge before the course, and needs to have the 'after'the course' comparison.

Text in a lot of figures schematics, pie charts, etc is very small. Please make much larger.
Could combine sonme figures onto a single page.